# Nonlinear Methods Most Applied to Heart-Rate Time Series: A Review

**DOI:** 10.3390/e22030309

**Published:** 2020-03-09

**Authors:** Teresa Henriques, Maria Ribeiro, Andreia Teixeira, Luísa Castro, Luís Antunes, Cristina Costa-Santos

**Affiliations:** 1Centre for Health Technology and Services Research (CINTESIS), Faculty of Medicine University of Porto, 4200-450 Porto, Portugal; andreiasofiat@med.up.pt (A.T.); luisa.castro@fc.up.pt (L.C.); csantos@med.up.pt (C.C.-S.); 2Health Information and Decision Sciences Department-MEDCIDS, Faculty of Medicine, University of Porto, 4200-450 Porto, Portugal; 3Institute for Systems and Computer Engineering, Technology and Science (INESC-TEC), 4200-465 Porto, Portugal; maria.r.ribeiro@inesctec.pt (M.R.); lfa@fc.up.pt (L.A.); 4Computer Science Department, Faculty of Sciences, University of Porto, 4169-007 Porto, Portugal

**Keywords:** nonlinear methods, heart-rate dynamics, time series

## Abstract

The heart-rate dynamics are one of the most analyzed physiological interactions. Many mathematical methods were proposed to evaluate heart-rate variability. These methods have been successfully applied in research to expand knowledge concerning the cardiovascular dynamics in healthy as well as in pathological conditions. Notwithstanding, they are still far from clinical practice. In this paper, we aim to review the nonlinear methods most used to assess heart-rate dynamics. We focused on methods based on concepts of chaos, fractality, and complexity: Poincaré plot, recurrence plot analysis, fractal dimension (and the correlation dimension), detrended fluctuation analysis, Hurst exponent, Lyapunov exponent entropies (Shannon, conditional, approximate, sample entropy, and multiscale entropy), and symbolic dynamics. We present the description of the methods along with their most notable applications.

## 1. Introduction

The term chaos, in science, refers to a mathematical approach dealing with systems that are fully describable but which generate randomly appearing outputs under certain conditions [1]. Chaos theory deals with patterns in the time evolution of a nonlinear system that is sensitive to initial conditions. On the other hand, a fractal is an object composed of subunits that resemble the larger-scale structure [2]. This property of self-similarity (or scale invariance) means that the details of the structures are similar but not necessarily identical when zooming at different resolutions. A fractal organization is flexible, and the breakdown of this scale invariance may lead to a more rigid and less adaptable system with either random or highly correlated structure. Nevertheless, the definition of fractal goes beyond self-similarity per se to include the idea of a detailed pattern repeating itself at increasingly small scales. Although fractals are irregular, not all irregular time series are fractal. The self-similarity of the system’s fluctuations can be observed when a signal is analyzed over different time scales.

A concept closely related to dynamical chaos and fractality is one of complexity. There are many different definitions of dynamical complexity. One of the most consensual is that complexity is a property of every system that quantifies the amount of structured information. Shannon demonstrated how the information within a signal could be quantified with absolute precision as the amount of unexpected data contained in the message (designated entropy) [3]. Kolmogorov proposed a definition of complexity that quantifies information on individual objects as the size of its smallest representation [4]. The Shannon information theory measures the average information from a random source measuring the randomness, unlike Kolmogorov complexity that presents a form of absolute information [5]. Furthermore, a variety of entropy measures using different estimators has been probed. Researches related to complexity are spread across different scientific fields, such as physics and biology, economics, and psychology. Most of these investigations are aimed at finding patterns, regularities, or laws that govern the dynamics of a data set [6]. Many nonlinear methods based on concepts of chaos, fractality, and complexity have been used in evaluating heart-rate variability to understand cardiovascular dynamics in healthy as well as in pathological conditions.

In past years, the scientific community published some review papers related to the nonlinear methods applied to heart-rate time series. In 1996, Mansier et al. [7] considered that the most widely used nonlinear methods of heart rate were the correlation dimension, the Lyapunov exponents, and the approximate entropy. The authors applied these methods to mouse heart-rate data before and after a single atropine injection. They concluded that the dose results in increased complexity of an assumed heart-rate attractor. Voss’s review in 2009 [8] summarizes some of the heart-rate variability (HRV) indices derived from nonlinear and fractal dynamics (fractal, entropy, symbolic dynamics measures, and Poincaré plot representation). It shows the relevance of the methods in clinical research, and it reflects on essential aspects related to their practical applications, although giving neither a mathematical description of algorithms nor the relations between methods. In 2016, Godoy [9] presented a literature review of linear and nonlinear methods to predict cardiovascular disease and concluded that nonlinear methods are more efficient than the linear ones. However, the paper lacks presenting the mathematical description of the algorithms as well as the relations between methods. In 2018, the review by Nayak et al. [10] dealt with the theory and recent applications of the nonlinear electrocardiogram signal analysis methods. In 2019, the review of García-Martínez et al. [11] analyzed nonlinear methods applied to electroencephalographies (EEG) signal analysis for emotion recognition. The last three reviews discussed included neither the representation methods (Poincaré plot and recurrence plot analyses) nor the symbolic dynamics. Furthermore, they did not refer to the limitations and advantages of each technique.

Most researchers, special the clinical ones, have difficulty with using the correct technique, with the correct parameters on the right type of data and interpreting the results accurately. Many of the HRV methods have a large number of cautions and limitations, especially when dealing with real-world data that can provide outcomes challenging to interpret or even completely wrong. We aim with this article to clarify some of those concepts and to help other researchers to apply them correctly.

In this article, the main goal, similar to the previous reviews, is to review the nonlinear methods used to assess heart-rate dynamics. The methods described in detailed are Poincaré plot (Section 3.2), recurrence plot analysis (Section 3.3), fractal dimension (Section 3.4), correlation dimension (Section 3.4.1), detrended fluctuation analysis (Section 3.5), Hurst exponent (Section 3.6), Lyapunov exponent (Section 3.7), entropies (Section 3.8), and symbolic dynamics (Section 3.9). Furthermore, we probe the relations between methods and summarize the most cited papers applying each measure to understanding the applicability of methods. We start this review by briefly explaining the heart-rate (HR) dynamics (Section 2). Then, in the nonlinear section, we will present some notation, followed by a description of each of the methods. We finish this review by giving, for each method, a summary of the most cited papers applying that measure to the HR time series.

## 2. Heart-Rate Dynamics

The electrocardiogram (ECG) registers the electrical impulses generated by the polarization and depolarization of cardiac tissue and translates them into a waveform used to measure the rate and regularity of the cardiac cycle (heartbeat). A typical ECG trace consists of a P wave, a QRS complex, a T wave, and a U wave (Figure 1).

The focus in this paper will be in the QRS complex, in particular, in the RR interval time series—a set of the time intervals between two consecutive R peaks. The R peaks are often classified as normal (N) or abnormal. Many studies use the normal to normal (NN) intervals time series instead of the RR ones, meaning that the time series used are the set of time differences between two consecutive R peaks, which are classified as normal. Numerous methods can be used to detect the R peaks. One of the most used ones is the Pan–Tompkins algorithm [12]. However, that topic will not be explored in this work. Please see Physionet for more details [13].

The regulatory system of the heart rate depends on many other physiological systems and especially the interaction of those systems. In this work, we will focus on the work developed to analyze only the heart-rate dynamics through the analysis of the RR (or NN) time series.

## 3. Nonlinear Methods

The nonlinear methods presented in this section are displayed in an organized form in Figure 2.

We start this section by giving some notation before describing each method in a detailed manner. The methods’ definitions are fundamental yet not more than understanding their applicability and how they relate with each other. Therefore, in each of the following sections, we also describe the most cited papers applying each measure to the HR time series. The methodology used to select the papers was the following:Pubmed search: all the papers that contained the method’s name *AND ((“Heart Rate” [Mesh] OR “Heart Rate Fetal” [Mesh]) OR (“Cardiotocography” [Mesh]) OR (“Electrocardiography” [Mesh])) AND (humans[MeSH Terms])*. The query details are described in Appendix D.the number of citations in Google Scholar was accessed for each paper.the five most cited papers for each method were selected.

The number of papers published in the last 20 years (from 1997 to 2017) for each measure (or group of measures) are displayed in Figure 3. Note that we did not use the 2018 information once we considered that those values were incomplete at the time of the search. The results on the left panel show that, although in the early years the fractal dimension (FD), the Poincaré plot (Pplot), and the approximate entropy (ApEn) were the most used methods, in the most recent years, the most applied methods to HR are the Pplot, the sample entropy (SampEn), and the multiscale entropy (MSE), followed by the detrended fluctuation analysis (DFA) and the ApEn. Moreover, the right panel of Figure 3 exhibits a tendency to reach higher values of the number of papers applying the method 10/20 years after the method’s proposal. This plot emphasizes the conclusions of the previous one that the SampEn and the Pplot are two of the most applied methods. Note that, for some methods, there are no papers published immediately after the method was proposed. This might be due to several reasons, such as the method was just applied to the HR time series some years after the proposal or we had not found papers available online before 1975.

### 3.1. Notation

A time series, as a set of *N* consecutive data points, is defined as X={x(i),i=1,…,N}, where, in this case, each data point will represent the value of an RR (or NN) interval. From the original time series *X*, let us define the vectors $X_{m}^{\tau }\left(i\right)$ as follow:(1)$$X_{m}^{\tau }\left(i\right)=\left(x\left(i\right),x\left(i+\tau \right),x\left(i+2\tau \right),\ldots ,x\left(i+\left(m-1\right)*\tau \right)\right)$$
with i=1,…,K, where K=[N−(m−1)∗τ], *m* is the embedding dimension and τ is the embedding lag. The choice of appropriate embedding parameters is critical. Appendix A presents two subsections that briefly describe some approaches used to estimate these parameters (Section A.1: Estimation of minimum embedding dimension (m) and Section A.2: time delay embedding estimation (τ)).

Note that ln will be used to represent the natural logarithm of a number.

### 3.2. Poincaré Plot

#### 3.2.1. Description of Poincaré Plot

The Poincaré plot (Pplot), also known as a return or delay map, allows assessing the heartbeat dynamics based on a simplified phase-space embedding. The Pplot is a two-dimensional graphic (scatter plot) in which each RR interval, x(i), is plotted as a function of the previous RR interval, x(i−1). The Pplot analysis is an emerging quantitative-visual technique, whereby the shape of the plot provides summary information on the behavior of the heart [14,15]. For a healthy heart, the cloud of points presents a comet shape oriented along the line of identity; the cardiac heart-failure dynamics are characterized by a stretched elliptical-shaped cloud of points also along the line of identity. In the atrial fibrillation (AF) case, the cloud of points presents a more circular shape, similar to what happens with the white noise time series (see Figure 4).

A number of techniques were developed attempting to summarize the plot’s geometric appearance quantitatively. The geometrical descriptors, as the ellipse fitting technique, the histogram techniques, and the correlation coefficient, are the most popular in the clinical and HRV literature [16]. The distribution of points along the perpendicular to the line of identity reflects the level of short-term variability measured by SD1 [15,17,18]. On the other hand, the dispersion of points along the line-of-identity is thought to indicate the level of long-term variability and is measured by the standard deviation denoted by SD2 [19].

The standard deviation of the RR intervals, denoted by SDRR, is often employed as a measure of overall HRV. It is defined as the square root of the variance of the RR intervals,
(2)$$SDRR=\sqrt{E\left[X^{2}\right]-E{\left[X\right]}^{2}}$$
where E[X] is the mean of the RR interval. The standard deviation of the successive differences of the RR intervals, denoted by SDSD, is an important measure of short-term HRV. It is defined as the square root of the variance of the sequence ΔX(i)=x(i)−x(i+1),
(3)$$SDSD=\sqrt{E\left[\Delta X^{2}\right]-E{\left[\Delta X\right]}^{2}}.$$

Note that E[ΔX]=0 for stationary intervals; therefore, SDSD is equivalent to the root mean square of the successive differences, denoted by RMSSD.

The geometric indices obtained by fitting an ellipse to the Pplot are dependent on the general time-domain HRV indices. The width of the Pplot is a linear scaling of the most common statistic used to measure short-term HRV, the SDSD index. In fact, the width of the Pplot correlates with other measures of short-term HRV [15,20].
(4)$$SD1^{2}=Var\left(\frac{1}{\sqrt{2}}X\left(i\right)-\frac{1}{\sqrt{2}}X\left(i+1\right)\right)=\frac{1}{2}Var\left(\Delta X\right)=\frac{1}{2}SDSD^{2}$$
where Var represents the variance.
(5)$$SD2^{2}=2SDRR^{2}-\frac{1}{2}SDSD^{2}$$

Two simple generalizations of the Pplot—lagged Poincaré plots and higher-order Poincaré plots—can also be encountered in the literature. In lagged Poincaré plots (lag−c), x(i) is plotted against x(i+c), where *c* is some small positive integer value. In general, the plot is still clustered around the line of identity. However, the length and width of the plot are altered as the lag is increased. On the other hand, considering the standard Pplot to be of the first-order, the second-order Pplot is a 3D scatterplot of the triples (x(i),x(i+1),x(i+2)). There are three orthogonal views of the shape of this plot, resulting in 2D projections onto each of the coordinate planes (x(i),x(i+1)),(x(i+1),x(i+2)) and (x(i),x(i+2)). The first two views are equivalent to the standard Pplot, and the third is the lag-2 Pplot. This idea can be extended into higher dimensions, with the projections of the plot onto coordinate planes being lagged Poincaré plots. Therefore, an order *c* Pplot is geometrically described by the set of lagged Poincaré plots up to and including lag-*c* [16].

The Pplot is a powerful tool not only for graphically representing the summary statistics but also for beat-to-beat structure. Nongeometric techniques, such as scanning parameters [19,21,22,23,24] and image distribution measures [25], are likely to be measuring independent, nonlinear information on the intervals. However, they are not nearly as popular as the “linear” Pplot measures in the literature [16].

#### 3.2.2. Applications of Poincaré Plot

The Pplot is one of the methods presented in this paper most used in the HR field. More than 300 papers have been published applying this method to the HR time series, creating more than eighteen thousand citations. The five most cited papers that applied Pplot to HR time series are References [19,20,26,27,28]. Tulppo et al. [19] studied the beat-to-beat HR dynamics for 10 subjects during incremental doses of atropine followed by exercise and during exercise without an autonomic block for 31 subjects. Poincaré quantitative measures were compared with linear measures of HRV and ApEn at rest and during exercise. The SD1/SD2 ratio had a modest correlation with ApEn at rest (r=−0.69, *p*-values <0.001), but not during exercise (r=0.27, nonsignificant *p*-value). All measures of vagal modulation of HR decreased progressively until the ventilatory threshold level was reached when sympathetic activation was reflected as changes in the SD1/SD2 ratio. Brennan et al. [20] analyzed if existing measures of Poincaré plot geometry—converting the two-dimensional plot into various one-dimensional views by fitting an ellipse to the plot shape and measuring the correlation coefficient of the plot—reflect nonlinear features of HRV. They showed that they all measure linear aspects of the RR intervals as other existing HRV indexes already do. The authors conclude that the problem with the summary methods of quantifying the Pplot is that they ignore the important beat-to-beat structure displayed by the plot. In Reference [26], the fast Fourier transform and Pplot were applied to compare RR intervals times series obtained from the Polar S810 heart-rate monitor (HRM) with an ECG during an orthostatic test of 18 healthy men. They concluded that narrow correlation limits, good correlations, and small effect sizes support the validity of the Polar S810 HRM to measure RR intervals and did the subsequent analysis of HRV in the supine position. However, caution should be taken in the standing position for parameters sensitive to short-term variability (RMSSD and SD1). Babloyantz and Destexhe [27] used the Pplot, power spectrum, the autocorrelation function, the phase portrait, correlation dimension (CD), Lyapunov exponent (LE), and Kolmogorov entropy to study cardiac activity of 36 ECG-leads recorded from 4 healthy resting individuals. All measures point to the fact that the normal heart is not a perfect oscillator. Tulppo et al. [28] studied the effects of age and physical fitness on vagal modulation of heart rate during exercise by analyzing the RR intervals variability from Pplot (SD1) at rest and different phases of a bicycle exercise test in 110 healthy males. SD1 was higher at rest in the young subjects than in the middle-aged or old subjects (39±14, 27±16, and 21±8, respectively), but the age-related differences in SD1 were smaller during exercise. The age-matched subjects with good, average, and poor physical fitness showed no difference in SD1 at rest (32±17, 28±13, and 26±11, respectively), but SD1 differed significantly among the groups from a low to a moderate exercise intensity level (13±6, 10±5, and 6±3 for good, average, and poor fitness group, respectively).

### 3.3. Recurrence Plot Analysis

#### 3.3.1. Description of Recurrence Plot Analysis

The recurrence plot (RP) method was introduced in 1987 by Eckmann et al. [29] to visualize the recurrence of dynamical systems in phase space. In this method, a K×K recurrence matrix (RM) is constructed, where the matrix of which elements $RM_{i,j}$ are defined as follows:(6)$$RM_{i,j}=\Theta \left(r-d\left(X_{m}^{\tau }\left(i\right),X_{m}^{\tau }\left(j\right)\right)\right)$$
with i,j=1,…,K, where *r* is a threshold distance, *d* is the Euclidean distance, Θ(.) is the Heaviside function, and τ is the embedding lag (Section A.2).

The Euclidean distance is defined as follows:(7)$$d\left(X_{m}^{1}\left(i\right),X_{m}^{1}\left(j\right)\right)=\sqrt{\sum_{k=1}^{m}{\left(X_{m}^{1}\left(i,k\right)-X_{m}^{1}\left(j,k\right)\right)}^{2}}$$
where $X_{m}^{1}\left(i,k\right)$ and $X_{m}^{1}\left(j,k\right)$ refer to the *k*th element of the series $X_{m}^{1}\left(i\right)$ and $X_{m}^{1}\left(j\right)$, respectively. If two-phase space vectors $X_{m}^{\tau }\left(i\right)$ and $X_{m}^{\tau }\left(j\right)$ are sufficiently close together, then $RM_{i,j}=1$; otherwise $RM_{i,j}$ is 0. The RP is the representation of the matrix RM as a black (for ones) and white (for zeros) image.

A crucial parameter of an RP is the threshold distance, *r*. If *r* is too small, there may be almost no recurrence points, and we cannot learn anything about the recurrence structure of the underlying system. On the other hand, if *r* is chosen too large, almost every point is a “neighbor” of every other data point, which leads to a lot of artifacts [30]. Several criteria for the choice of the distance threshold *r* have been proposed: a few percents of the maximum phase space diameter [31], a value which should not exceed 10% of the mean or the maximum phase space diameter [32,33], or a value that ensures a recurrence point density of approximately 1% [34]. Another approach is to choose *r* according to the recurrence point density of the RP by seeking a scaling region in the recurrence point density [34] or by taking into account that a measurement of a process is a composition of the real signal and some observational noise with standard deviation [35]. One of the most used approaches uses a fixed number of neighbors for every point of the trajectory, called the fixed amount of nearest neighbors [29]. In this approach, $r_{i}$ changes for each state $X_{m}^{\tau }\left(i\right)$ to ensure that all columns of the RP have the same recurrence density. Using this neighborhood criterion $r_{i}$ can be adjusted in such a way that the recurrence rate has a fixed, predetermined value [30].

Several measures known as recurrence quantification analysis (RQA) were developed to quantify different properties of the temporal evolution of a system, such as stability, complexity, and the occurrence of epochs of chaos vs. order in the behavior of the system [32,36,37,38]. These measures are based on the recurrence point density and the diagonal and vertical line structures of the RP. The diagonals reflect the repetitive occurrence of similar sequences of states in the system dynamics and express the similarity of system behavior in two distinct time sequences. The vertical lines result from the persistence of one state during some time interval. The parameters extracted from both diagonal and vertical lines are briefly described in the Appendix B.

#### 3.3.2. Applications of Recurrence Plot Analysis

Papers exploiting recurrence plots applied to the HR time series have close to three thousand citations. Melillo et al. [39] probed RP, Pplot, ApEn, CD, and DFA to study 42 students under stress due to university examination. The features’ maximum line length in RP, SD2, and short-term fractal scaling exponent ($\alpha_{1}$) of the DFA were significantly reduced during university examination as compared with rest sessions. In contrast, the average line length in RP and recurrence rate increased significantly during stress. Acharya et al. [40] compared many methods, such as RP, Pplot, RQA, Shannon entropy (SE), ApEn, SampEn, DFA, and conditional entropy (CE), to analyze HR signals from both healthy and coronary artery disease (CAD) subjects, in time, frequency, and nonlinear domains. The authors found that RQA parameters were higher for patients with CAD compared with healthy individuals. The activity of subjects with CAD is lower than healthy ones, meaning similar signal patterns repeat more frequently. Konvalinka et al. [41] applied the RQA measures and cross-recurrence quantification analysis to compared synchronized arousal between performers and related spectators in a fire-walking ritual. They analyzed ECG and instantaneous HR from interbeat intervals from 38 participants. The methods identified similarities of arousal during the 30-min routine between fire-walkers and related spectators but not unrelated spectators. Zbilut et al. [42] gave two examples substantiating the utility of RQA in cardiovascular physiology, especially in circumstances where standard Fast Fourier Transform methods are found to be wanting. In the first example, the author concludes that the recurrence graph shows that the ECG signal is not continuous but characterized by singularities. The second example demonstrates the utility of windowed RQA to localize in time-specific events. Censi et al. [43] used the RP and RQA in time series of 19 subjects with chronic AF to determine the presence of organization of atrial activation processes during AF by analyzing whether the activation sequences are random or are governed by deterministic mechanisms. The RP measures detected transient recurrent patterns in all the episodes. The detection of recurrent spatiotemporal patterns along with the results of surrogate data indicates that, during AF, there is a certain degree of local organization.

### 3.4. Fractal Dimension

#### 3.4.1. Description of Fractal Dimension Methods

A fractal dimension (FD) is a statistical index of how details in a pattern change with the scale at which it is measured. The FD emerges to provide a measure of how much space an object occupies between Euclidean dimensions. The FD of a waveform represents a powerful tool for transient detection. The higher the FD, the more irregular the signal is, i.e., the more self-similar the signal will be.

From the several algorithms available to calculate the FD of a time series, the four most common are the correlation dimension, the box-counting dimension [44], and the algorithms proposed by Katz [45] and by Higuchi [46,47].

##### Correlation Dimension

The correlation dimension (CD), one of the most widely used measures of fractal dimension, can be considered as a measure for the number of independent variables needed to define the total system in phase space [48]. To analyze the complexity of a system, usually, a transition from the time-domain to the phase space is needed.

For each vector $X_{m}^{1}\left(i\right)$, the relative number of vectors $X_{m}^{1}\left(j\right)$ for which $d(X_{m}^{1}\left(i\right),X_{m}^{1}\left(j\right))\leq r$, where *r* is referred as a threshold tolerance value, is computed as follows:(8)$$C_{m}^{r}\left(i\right)=\frac{1}{N-m+1}\sum_{j=i+1}^{N-m+1}RM_{ij}$$
where $RM_{ij}$ was defined in Equation (Equation 6).

The probability that two chosen points are close to each other with a distance smaller than *r* is computed by averaging $C_{m}^{r}\left(i\right)$ over *i*:(9)$$C_{m}^{r}=\frac{2}{N-m}\sum_{i=1}^{N-m+1}C_{m}^{r}\left(i\right).$$

The $C_{m}^{r}$ index is computed for increasing values of the embedding dimensions *m* (usually, here, the embedding dimension varies between 2 and 30 [49]). Grassberger and Procaccia [50] showed that CD could be obtained from the following:(10)$$CD=\lim_{r\to 0}\lim_{N\to \infty }\frac{log(C_{m}^{r})}{log(r)}.$$

The slopes of the log-log (log(r) and $log(C_{m}^{r})$) plot are determined, obtaining a sequence of d(m). As *m* increases, d(m) tends to a constant value of saturation, which is the CD value [51]. In practice, this limit value is approximated by the slope of the regression curve $\left(log\left(r\right),log\left(C_{m}^{r}\right)\right)$ [52].

Another approach for estimating the CD value is the Levenberg–Marquardt method [53]. The exponential model for the d(m) values is $d\left(m\right)=CD\left(1-r^{-km}\right)$, where CD and *k* are the parameters of the curve, where the former represents the asymptotic value of the curve when m→∞ and the latter is the exponential constant.

##### Algorithm Proposed by Barabasi and Stanley

The main idea of the box-counting method is to analyze complex patterns by breaking the signal into smaller and smaller pieces, typically “box”-shaped, and by analyzing the pieces at each scale. The minimum number of elements of a given size (ε), necessary to fully cover the curve (*S*), of dimension *d* is counted ($N_{\varepsilon }$).
(11)$$N_{\varepsilon }\left(S\right)∼\frac{1}{\varepsilon^{d}}as\varepsilon \to 0$$

As the size of the element approaches zero, the total area covered by the area elements will converge to the measure of the curve. This way, the $FD_{B}$ can be estimated via a box-counting algorithm as proposed by Barabasi and Stanley [44] as follows:(12)$$FD_{B}=-lim_{\varepsilon \to 0}\frac{ln(N_{\varepsilon }\left(S\right))}{ln(\varepsilon )}.$$

##### Algorithm Proposed by Katz

The fractal dimension ($FD_{K}$) of the waveform representing the time series is estimated using Katz method [45] as follows:(13)$$FD_{K}=\frac{log(L)}{log(d)}$$
where *L* is the total length of the curve calculated as the sum of the distance between the successive data points and *d* is the diameter or planar extent of the curve, estimated as the distance between the first point and the point in the sequence that gives the farthest distance. For the signals that do not cross themselves, it can be expressed as
(14)$$d=max_{i=2,\ldots ,N}\left|x\left(1\right)-x\left(i\right)\right|.$$

##### Algorithm Proposed by Higuchi

Higuchi’s method [46,47] is a very efficient algorithm to calculate the FD of a curve, and it has been increasingly used for the analysis of time series. For a time series expressed by x(i),i=1,…,N, new series $Y_{r}^{m}$ are obtained as follow:(15)$$Y_{r}^{m}=\left(x\left(m\right),x\left(m+r\right),x\left(m+2r\right),\ldots ,x\left(m+\left\lfloor \frac{\left(N-m\right)}{r}\right\rfloor r\right)\right),m=1,\ldots ,r$$
where ⌊.⌋ denotes the Gauss notation and where *m* and *r* are integers that indicate the initial time and the time interval, respectively.

The length of the new series $Y_{r}^{m}$, $L_{m}\left(r\right)$ is defined as follows:(16)$$L_{m}\left(r\right)=\left[\left(\sum_{i=1}^{\left[\frac{N-m}{r}\right]}\left|x\left(m+ir\right)-x\left(m+\left(i-1\right)r\right)\right|\right)\times \frac{N-1}{\left\lfloor \frac{\left(N-m\right)}{r}\right\rfloor r}\right]\frac{1}{r}.$$

The length of the L(r) for the time interval *r* is obtained by averaging all the subseries lengths $L_{m}\left(r\right)$ that have been obtained for a given *r* value.

If L(r) is proportional to $r^{-D}$, the curve describing the shape is fractal-like with the dimension *D*. Thus, if L(r) is plotted against *r*, on a double logarithmic scale $\left(ln\left(\frac{1}{r}\right),ln\left(L\left(r\right)\right)\right)$, the points should fall on a straight line with a slope equal to −D. The coefficient of the linear regression of the plot is taken as an estimate of the FD of the epoch. Applying the above relation implies the proper choice of a maximum value of *r* for which the relationship between L(r) and $r^{-D}$ is approximately linear.

#### 3.4.2. Applications of Fractal Dimension

Since the first algorithm of fractal dimension was proposed, more than 120 papers have been published applying these algorithms to the HR time series, creating more than 4700 citations. Beckers et al. [54] aimed to compare the effect of gender, age, and the day–night variations applying several nonlinear indexes (FD—Katz algorithm, DFA, CD, LE, and ApEn) to RR intervals time series from a healthy population. Their findings are of most interest to this topic. The FD values were FD=1.27±0.09 (males day); FD=1.20±0.08 (males night); FD=1.28±0.08 (female day); and FD=1.22±0.08 (female night). The CD was computed using the algorithm described before [48]. The values obtained were CD=3.97±0.72 (males day); CD=4.37±1.30 (males night); CD=4.15±0.75 (female day); and CD=4.41±1.29 (female night). The authors found that (1) all nonlinear indices present a day–night variation except for the CD in the female population; (2) gender-related differences only existed in ApEn, DFA, and the LE; (3) all nonlinear indexes were significantly correlated with age during daytime hours. During the night, the relation with age disappeared in some indexes, such as DFA. (4) The FD was not related to linear HRV indexes but was correlated with every other nonlinear index except for the CD; and (5) the CD only moderately correlated with the proportion of high frequencies power and also showed positive correlations with ApEn and negative correlation with DFA. As referenced before (Section 3.2.2), Babloyantz and Destexhe [27] used the CD in RR intervals time series from 4 healthy subjects obtaining values around 5.9±0.4. Owis et al. [55] successfully used the largest Lyapunov exponent (LLE) and the CD in RR intervals time series from five different signal types from the MIT-BIH (Massachusetts Institute of Technology - Boston’s Beth Israel Hospital) Arrhythmia Database: normal, ventricular couplet, ventricular tachycardia, ventricular bigeminy, and ventricular fibrillation. The CD (computed using the Grassberger and Procaccia algorithm) was significantly higher in the normal group (CD=3.27±0.42) when compared with each of the pathological ones. Nakamura et al. [56] studied the change of FD during physical activity, probing the RR time series of 10 subjects. The FD index was computed using the power-law component β. Most of the calculated values for β were between 1 and 3 (indicating fractal dynamics); therefore, the FD index was computed as FD=1/(β−1) for 1<β≤3. With the increment of exercise intensity, a decrease in parasympathetic indicator was accompanied by a decrease in the FD. During mild exercise, values around 3 that decreased to lower than 2 were shown. Vaughn et al. [57] analyzed the HRV in normal adults during sleep using Katz’s FD index. They found that FD was significantly different across sleep stage with values of FD=2.04±0.22 (awake); FD=2.87±0.80 (stage 2); FD=2.16±0.59 (stage 3/4); and FD=2.43±0.40 (rapid eye movement (REM)). Finally, Turcott et al. [58] compared the FD computed using the box-counting algorithm in 15 RR intervals time series from healthy subjects and 15 time series from heart-failure patients. They showed that the healthy patients had values of FD of 2.75±0.20 and that the heart failure patients presented significantly lower values (FD=2.17±0.29).

### 3.5. Detrended Fluctuation Analysis

#### 3.5.1. Description of Detrended Fluctuation Analysis

Detrended fluctuation analysis (DFA) quantifies intrinsic fractal-like (short and long-range) correlation properties of dynamic systems [59]. This technique is a modification of root mean square analysis of random walks applied to nonstationary signals [60].

First, the time series (of length *N*) is integrated. Then, the integrated time series is divided into $N_{n}$ windows of equal length *n*. In each window of length *n*, a least-squares line is fitted to the data. The y-coordinate of the straight-line segments is denoted by $y_{n}\left(k\right)$. Next, the integrated time series is detrended, $y_{n}\left(k\right)$, in each window. The root mean square fluctuation of this integrated and detrended series is calculated using the following equation:(17)$$F\left(n\right)=\sqrt{\frac{1}{N}\sum_{k=1}^{N}{\left[y\left(k\right)-y_{n}\left(k\right)\right]}^{2}}.$$

This computation is repeated over all time scales (box sizes) to characterize the relationship between F(n), the average fluctuation, and the box size, *n*. Typically, F(n) increases with window size, according to $F\left(n\right)\propto n^{\alpha }$. The α exponent can be viewed as an indicator of the “roughness” of the original time series: the larger the value of α, the smoother the time series:if α≃0.5, the time series represents uncorrelated randomness (white noise);if α≃1 (1/f-noise), the time series has long-range correlations and exhibits scale-invariant properties;if α≃1.5, the time series represents a random walk (Brownian motion).

From this perspective, 1/f-noise can be interpreted as a compromise between the unpredictability of white noise and the Brownian noise motion [61].

Usually, the DFA method involves the estimation of a short-term fractal scaling exponent, $\alpha_{1}$, and a long-term scaling exponent, $\alpha_{2}$.

DFA as such is a monofractal method, but the multifractal analysis also exists. The multifractal analysis describes signals that are more complex than those fully characterized by a monofractal model but requires many local and theoretically infinite exponents to characterize their scaling properties completely. See Appendix C to more details about multifractal DFA.

#### 3.5.2. Applications of Detrended Fluctuation Analysis

Since 1994, when the DFA measure was proposed, more than 200 papers have been published applying this method to the HR time series and creating more than fourteen thousand citations. Peng et al. [62] applied the DFA to 24-h RR intervals time series from 12 healthy adults and 15 adults with severe heart failure. The α exponents, $\alpha_{1}$ and $\alpha_{2}$, were calculated for segments of 2 h. They found significantly lower $\alpha_{1}$ and higher $\alpha_{2}$ values for the group of congestive heart failure subjects. Iyengar et al. [63] tested the alterations of fractal correlations in sinus rhythm interbeat interval dynamics with aging. Continuous ECG signals were recorded for 120 min while subjects lay supine. The RR intervals time series were analyzed without detrending. Three exponents were explored: a short-range scaling exponent, $\alpha_{s}$, over periods of 4 to $n_{b}p$ beats (where $n_{b}p$ usually ranged from 25 to 30 heartbeats), and a long-range exponent, $\alpha_{1}$, over longer periods. A β exponent was also considered as β=2α−1. α and β were found to be significantly lower in the young group, whether the $\alpha_{1}$ was higher in the same group. Ho et al. [64] computed the DFA and the ApEn in 2-h ambulatory ECG recordings. The NN intervals time series were analyzed for 28 subjects with CHF and 41 healthy controls. Over a mean follow-up period of 1.9 years, 12 deaths occurred (9 CHF patients and 3 healthy subjects). The authors concluded that “patients with CHF show a breakdown of the physiological long-range (or fractal) correlations of HR, as quantified by DFA, and the DFA index is a significant prognostic indicator that may complement traditional HRV measures in assessing risk." Bunde and colleagues [65] investigated the heart rhythm within the different sleep stages: deep sleep, light sleep, and REM sleep. Thirty RR intervals time series from 15 healthy individuals and 47 RR intervals time series from 26 individuals suffering from moderate sleep apnea with an average length of 7.5 h were analyzed. In deep and light-sleep, there was a loss of correlations, while in REM sleep, long-range correlations were observed with an α≈0.85. Similar findings were obtained for all healthy subjects and sleep apnea patients. Lastly, Pikkjämsä et al. [66] also investigated the effects of age in the RR interval from 24-h ECG signals in 104 healthy subjects. The subjects were divided into four groups: (1) 27 children, <15 years old; (2) 29 young adults, 15 to 40 years old; (3) 29 middle-aged, 40 to 60 years old; and (4) 29 elderly, >60 years old. They computed the DFA exponents—$\alpha_{1}$, the short-term (4 to 11 beats), and $\alpha_{2}$, the intermediate-term (>11 beats)—and ApEn in segments of 8000 heartbeats, averaged to assess the 24 h, and in segments of 4000 heartbeats for assessing the night and day time periods. The main results were that (1) both children and young adults showed RR interval dynamics with 1/f behavior ($\alpha_{1}$ and $\alpha_{2}\approx 1.0$); (2) a decrease of ApEn and an increase of $\alpha_{2}$ (to $1/f^{2}$ behavior) were observed with increasing age; (3) increase of ApEn and decrease of $\alpha_{1}$ during nighttime indicate increased variance and complexity of HR dynamics at night; and (4) women had significantly lower $\alpha_{1}$ (closer to 1) than men.

### 3.6. Hurst Exponent

#### 3.6.1. Description of Hurst Exponent

Initially defined by Harold Edwin Hurst [67,68] to develop a law for regularities of the Nile water level, the Hurst exponent (HE) is a dimensionless estimator used to evaluate the self-similarity and the long-range correlation properties of time series. Unlike other nonlinear dynamic features, the HE does not involve state space reconstruction. There is a straightforward relationship between the HE and the fractal dimension FD, given by FD=E+1−HE, where *E* is the Euclidean dimension, which for time series is 1 obtaining their relationship FD=2−HE [69].

The oldest description of the HE is defined in terms of the asymptotic behavior of the rescaled range (a statistical measure of the variability of a time series) as a function of the period of a time series as follows:(18)$$E\left[\frac{R(N)}{S(N)}\right]=c\cdot N^{HE}asN\to \infty$$
where S(N) is the standard deviation, *c* is an arbitrary constant, and the range R(N) is defined as the difference between the maximum and the minimum values of a given time series.

There are many algorithms to estimate the HE parameter. The most immediate one is derived from the definition. First, the time series of length *N* is subdivided into segments of length *T*. Then, the ratio R/S is computed for each segment and the average, for all segment, is calculated. These steps are repeated for several values of *T*. The HE can also be computed using the periodogram ($P_{xx}$), which is an approximation of the power spectral density (PSD) [70]. The Bartlett’s method [71] (the method of averaged periodograms [72]) or its modification, the Welch method [73], are the most used method for, in practice, estimating power spectral density. As in the previous method, the time series of length *N* is divided into segments of length *T*. In Bartlett’s method, for each segment, the periodogram is computed using the discrete Fourier transform, then the squared magnitude of the result is calculated and divided by *T*. Average the result of the periodograms above for the data segments. The averaging reduces the variance, compared to the original *N* point data segment [71,74]. The Welch method differs since it allows the segments of the time series of each periodogram to overlap. Once the PSD is estimated, it is possible to determine β parameter as the slope of the straight line fitted using least squares.

Moreover, the computation of HE index is straightforward using the following relation:(19)β=1+2HE.

Another increasingly used approach is related to the DFA method (described in the previous section). The α exponent is related to the HE parameter by [75] α=1+HE. The advantages of DFA over conventional methods (periodogram and R/S method) are that of permitting the detection of long-range correlations in time series with non-stationarities and of avoiding the spurious detection of apparent long-range correlations that are an artifact of non-stationarity [60,62].

The HE may range between 0 and 1 and can indicate the following:

0<HE<0.5: time series has long-range anti-correlations;

HE=0.5: there is no correlation in the time series;

0.5<HE<1: there are long-range correlations in the time series.

HE=1: the time series is defined self-similar, i.e., it has a perfect correlation between increments. However, the literature on HRV conventionally uses the term self-similarity even if HE differs from 1 [76].

#### 3.6.2. Applications of Hurst Exponent

The Hurst exponent is one of the methods presented in this paper that is less used in the HR field. Yamamoto et al. [77] applied it with success in the RR intervals time series of 10 healthy vs. 11 cardiac patients. Cardiac disease patients presented a higher HE than the healthy ones. This paper has more than 100 citations. In Reference [78], the HE was applied to 179 RR intervals time series of 65 young volunteers and 114 patients with CAD. The HE was found to be highly correlated with the Poincaré ratio SD1/SD2 (*r* = 0.92 in patients with CAD and *r* = −0.96 in the healthy volunteers). As mentioned in the description of this method, the relationship with the DFA has been probed. Struzik et al. [79] compared the RR intervals time series of 115 healthy subjects with 12 patients with CHF using the HE as the first order of DFA. They found that the parasympathetic suppression by CHF results in an increase in the HE from 1/f range (HE≈0.09 for healthy subjects) towards random walk scaling $1/f^{2}$ (HE>0.2). Martinis et al. [80] studied the change in the HE of RR intervals during physical activities in healthy subjects and patients with stable angina pectoris. The HE values higher than 0.5 showed the presence of long-time correlations in both groups. Differences were found, being higher in the healthy group, when the subjects were under specific physical activities. Acharya et al. [81] used the ApEn, FD, CD, LLE, and HE to detect sleep apnea from the ECG. All the nonlinear indices showed significant different mean values for apnoea, hypopnoea, and normal breathing ECG signals. The mean HE for normal breathing was 0.62±0.02 and lower close to 0.2, which means that RR intervals are negatively correlated.

### 3.7. Lyapunov Exponent

#### 3.7.1. Description of Lyapunov Exponent

The Lyapunov exponent (LE) is a measure of the system dependency on the initial conditions but also quantifies the predictability of the system [82]. A system embedded in a *m*-dimensional phase space has *m* LE. The LE, λ, is a measure of the exponential divergence (λ > 0) or convergence (λ < 0). The value of LE increases, corresponding to lower predictability, as the degree of chaos becomes higher; a positive LE is a strong indicator of chaos [83,84,85]; therefore, the computation only of the largest Lyapunov exponent (LLE) is sufficient to assess chaos. The LLE is most commonly used in nonlinear analysis of physiological signals [86,87,88].

There are many algorithms available to estimate both LE and LLE [85,89,90,91,92,93,94]. The method proposed by Rosenstein et al. [91] is one of the most used since it is robust against data length. The algorithm looks for the nearest neighbor of each point on the trajectory. The distance between two neighboring points at instant 0 is defined by the following:(20)$$d_{i}\left(0\right)=min_{X_{m}^{\tau }\left(j\right)}d\left(X_{m}^{\tau }\left(j\right),X_{m}^{\tau }\left(i\right)\right)$$
where $d(X_{m}^{\tau }\left(j\right),X_{m}^{\tau }\left(i\right))$ is the Euclidean distance. This algorithm imposes a constraint that nearest neighbors are temporally separated at least by the mean period of the time series. The LLE is then estimated as the mean rate of separation of nearest neighbors, i.e., we can write
(21)$$d_{j}\left(i\right)\approx C_{j}\;e^{\lambda_{1}i\left(\Delta t\right)}$$
where $C_{j}$ is the initial separation. Taking logarithm on both sides, we obtain
(22)$$ln\left(d_{j}\left(i\right)\right)=ln\;C_{j}+\lambda_{1}i\left(\Delta t\right).$$
It represents a set of approximately parallel lines, where the slope is roughly proportional to the LLE. In practice, the LE is easily and accurately estimated using a least-squares Δt to the “average” line defined by the following:(23)$$y\left(n\right)=\frac{1}{\Delta t}\langle ln\;d_{i}\left(n\right)\rangle$$
where 〈·〉 denotes the average over all values of *i*. This last averaging step is the main feature that allows an accurate evaluation of λ even when we have a short and noisy data set.

Another widely used method was proposed by Wolf et al. [85] in 1985, where the calculation of the LLE is based on the average exponential rate of divergence or convergence of trajectories which are very close in phase space. Contrary to the Rosenstein method, in Wolf’s method, a single nearest neighbor is followed and repeatedly replaced when its separation from the initial trajectory grows beyond a certain threshold. However, this strategy does not take advantage of all the available data [91]. Another disadvantage of Wolf’s method is the requirement of an appropriate selection of parameters, not only the time delay and embedding dimension but also maximum and minimum scale parameters, an angular size, and trajectory evolution time [85,95].

#### 3.7.2. Applications of Lyapunov Exponent

In the work of Reference [27], discussed in Section 3.2.2, the authors found, for the 4 healthy resting individuals, a value of λ=0.38±0.08, which shows the presence of chaotic dynamics in HR signals. In Reference [55], briefly described before in Section 3.4.2, the authors use Wolf’s method to compute the LLE. The LLE was significantly lower in the normal group (λ=8.18±3.64) when compared with each of the pathological ones. Also, the LLE was the highest for ventricular couplet (λ=17.36±3.68) compared with all the other groups. In the Beckers et al. [54], already mentioned in Section 3.4.2, the LE main results were λ=0.27±0.07 (males day); λ=0.30±0.10 (males night); λ=0.25±0.06 (female day); and λ=0.28±0.09 (female night). They also found that the LE did not show any relation with the linear HRV indexes but showed a significant negative linear correlation with FD and a significant positive linear correlation with DFA. Acharya et al. [86] compared different age groups when applying the Pplot, the ApEn, the LLE, and the DFA to a set of 150 healthy subjects. They found LLE equal to λ=0.69±0.20 (5–15 years); λ=0.64±0.09 (15–35 years); λ=0.44±0.18 (35–55 years); and λ=0.44±0.29 (55–65 years). The results showed that LLE as well as the ApEn and the DFA significantly decrease with aging. Guzzeti et al. [49] compared the RR intervals time series from seven recently heart-transplanted patients and seven controls of similar age using the CD, Kolmogorov entropy, HE self-similarity exponent, and the LE. A reduction in all of the nonlinear indices was verified when comparing the transplanted patient group with the healthy one.

### 3.8. Entropies

#### 3.8.1. Description of Entropies Methods

##### Shannon Entropy

Shannon proposed the notion of entropy (Shannon entropy (SE)) to measure how the information within a signal can be quantified with absolute precision as the amount of unexpected data contained in the message [3,96]. The Shannon entropy is obtained by the following:(24)$$SE=-\sum_{i}p\left(x\left(i\right)\right)\cdot log\left(p(x(i))\right)$$
where p(x(i)) represents the probability of the point x(i).

##### Conditional Entropy

The conditional entropy (CE), also know as Kolmogorov–Sinay entropy [97], assesses the amount of information carried by the current RR sample when m−1 past samples of RR are known. The CE represents the difficulty in predicting future values based on past values of the same time series. When the future values of RR are completely predictable, given past values, the CE value is 0. If the past values of RR are not helpful to reduce the uncertainty associated with future RR values, the value of CE is equal to SE. In the approach introduced by Porta [98], the RR time series is recoded. Briefly, the RR series is first transformed into a sequence of symbols using a coarse-graining approach based on a uniform quantization procedure. The full range of the series was spread over ξ symbols, with a resolution given by $\frac{X_{max}-X_{min}}{\xi }$, where $X_{max}$ and $X_{min}$ are the maximum and the minimum of the series. After quantization, the RR series became a sequence $X^{\xi }=\left\{X^{\xi }\left(i\right),i=1,\ldots ,N\right\}$ of integer values ranging from 0 to ξ−1, where the $X^{\xi }$ series are transformed into a $\xi^{m}$ subseries $X_{m}^{\xi }\left(i\right)=\left(X^{\xi }\left(i\right),X^{\xi }\left(i-1\right),\ldots ,X^{\xi }\left(i-m+1\right)\right)$ with i=m,…,N, using the technique of the delayed coordinates [99]. The CE of RR is defined as follows:(25)$$CE^{m}=-\sum_{i=1}^{N-m+1}\left[p\left(X_{m}^{\xi }\left(i\right)\right)\cdot \sum_{i=2}^{N-m+1}\left[p\left(X^{\xi }\left(i\right)|X_{m-1}^{\xi }\left(i-1\right)\right)\cdot log\left(X^{\xi }\left(i\right)|X_{m-1}^{\xi }\left(i-1\right)\right)\right]\right]$$
where $p(X^{\xi }\left(i\right)|X_{m-1}^{\xi }\left(i-1\right))$ is the conditional probability of $X^{\xi }\left(i\right)$ given previous m−1 samples.

The concept of CE encompasses a wide range of entropy measures and estimates that have been proposed to quantify the complexity of a time series intended as the degree of predictability of the underlying process. These measures include corrected conditional entropy, approximate entropy, and sample entropy that are the estimators based, respectively, on the binning, kernel, and nearest neighbor [6]. Details of these three estimators are presented in the following.

##### Corrected Conditional Entropy

Since the percentage of patterns found only once grows monotonically towards 100% with *m*, CE always decreases toward 0 with *m* independently of the type of RR dynamics. In order to prevent the artificial decrease of the information carried by RR given *m* past samples, solely related to the shortness of the data sequence, the corrected conditional entropy (CCE) is defined as follows [98,100,101]:(26)$$CCE\left(m\right)=CE\left(m\right)+SE\left(1\right)\cdot perc\left(X_{m}^{\xi }\right).$$
where $X_{m}^{\xi }=\left\{X_{m}^{\xi }\left(i\right),i=1,\ldots ,N-m+1\right\}$ are the series of the patterns that can be constructed from $X^{\xi }$, $perc(X_{m}^{\xi })$ represents the fraction of patterns found only once in $X_{m}^{\xi }\;with\;0\leq perc\left(X_{m}^{\xi }\right)\leq 1$, and
(27)$$SE\left(m\right)=-\sum_{i}p\left(X_{m}^{\xi }\left(i\right)\right)\cdot log\left(p(X_{m}^{\xi }\left(i\right))\right)$$

The CCE(m) decreases towards 0 only in the case that RR is entirely predictable given past RR values; it remains constant when past RR values are not helpful to predict future RR, and it exhibits a minimum when past RR values are only partially helpful to predict RR. The minimum of $CCE,CCE_{min}$ represents the minimum amount of information carried by RR given its past values: the larger this value, the more significant the amount of information carried by RR and the smaller the predictability of RR based on its past values.

##### Approximate Entropy

The approximate entropy (ApEn), proposed by Pincus [102], exhibits good performance in the characterization of randomness even when the data sequences are not very long. In order to calculate the ApEn the new series of a vector of length *m*—embedding dimension—are constructed $X_{m}^{1}$. Similar to CD, for each vector $X_{m}^{1}\left(i\right)$, the value $C_{m}^{r}\left(i\right)$, where *r* is referred as a tolerance value, is computed as follows:(28)$$C_{m}^{r}\left(i\right)=\frac{1}{N-m+1}\sum_{j=1}^{N-m+1}RM_{ij}$$
where $RM_{ij}$ was defined in Equation (Equation 6) but where the distance function used is defined as follows:(29)$$d\left(X_{m}^{1}\left(i\right),X_{m}^{1}\left(j\right)\right)=max_{k=1,\ldots ,m}\left|x\left(i+k-1\right)-x\left(j+k-1\right)\right|.$$

Next, the average of the natural logarithm of $C_{m}^{r}\left(i\right)$ is computed for all *i*:(30)$$\Phi_{m}^{r}=\frac{1}{N-m+1}\sum_{i=1}^{N-m+1}ln\left(C_{m}^{r}\left(i\right)\right).$$

Since in practice *N* is a finite number, the statistical estimate is computed as follows:$$ApEn\left(m,r\right)=\left\{\begin{array}{cc} \Phi_{m}^{r}-\Phi_{m+1}^{r} & \;\mathrm{for}\;m>0\\ -\Phi_{1}^{r} & \;\mathrm{for}\;m=0. \end{array}\right.$$

The choice of the embedding dimension parameter *m* was already discussed in the beginning of this chapter. In the particular case of the ApEn, the most common value is m=2.

Regarding parameter *r*, several approaches are used. Pincus [102,103] recommends values between 10% and 25% of the standard deviation of the data, hence obtaining a scale-invariant measurement. The approach of choosing a fixed *r* value was also used with success [104,105]. However, the values of entropy in this case are usually highly correlated with the time series standard deviation. Lu et al. [106] showed that ApEn values varied significantly even within the defined range of *r* values and presented a new method for automatic selection of *r* that corresponds to the maximum ApEn value.

##### Sample Entropy

The sample entropy (SampEn) was introduced, with the same objective as ApEn, to evaluate the randomness of biological time series, in particular, the HR time series. The main limitation of the ApEn is the dependence on the record length. Meaning that the ApEn is lower for short records, and if one time series is higher than another, it should not remain higher for all conditions [107]. In order to overcome the limitations, the authors proposed a new family of statistics, SampEn(m,r), which, among other differences, eliminates self-matches, thereby reducing the computing time by one-half in comparison with ApEn. Therefore, the SampEn agrees with the theory for random numbers with known probabilistic character over a broad range of operating conditions. Also, due to the nonindependence of templates, it has a residual bias for very-short time series.

For the SampEn [107] calculation, the same parameters defined for ApEn, *m*, and *r* are required, considering *A* as the number of vector pairs of length m+1 having $d[X_{m}^{1}\left(i\right),X_{m}^{1}\left(j\right)]\leq r$, with i≠j and *B* as the total number of template matches of length *m* also with i≠j. The SampEn is defined as follows:(31)$$SampEn=-ln\frac{A}{B}.$$

##### Multiscale Entropy

Traditional entropy-based algorithms quantify the regularity of a time series. The multiscale entropy approach (MSE) [108,109] is inspired by Zhang’s proposal [110] and considers the information of a system’s dynamics on different time scales. The multiscale method can be separated into two parts. The first one is the construction of the time series scales: using the original signal, a scale, *s*, is created from the original time series, through a coarse-graining procedure, i.e., replacing *s* nonoverlapping points by their average. The second step concerns the calculation of the value of entropy for each time series scale. The most used entropies in this approach are the ApEn and the SampEn. The information of the different time scales is clustered in the complexity index defined as the area under the MSE curve obtained by plotting the entropy value as a function of scale [108,109].

However, the MSE algorithm described before presents some limitations [111]. More recently, several modifications of the MSE have been proposed [111,112,113,114]. Most of them suggest a different algorithm to create the time series scales, maintaining the second step, while other papers suggest the use of other entropies [115].

#### 3.8.2. Applications of Entropy Methods

##### 3.8.2.1. Applications of Shannon Entropy

The five top-cited papers applying SE to the HR time series are References [101,116,117,118,119]. Porta et al. [101] probed the SE, the CE, and classification of frequent deterministic patterns to the complexity analysis of short heart period variability series (300 cardiac beats) from 15 healthy young subjects during the sympathetic activation induced by head-up tilt and during the driving action produced by controlled respiration (10, 15, and 20 breaths/min—CR10, CR15, and CR20, respectively). The authors calculated the SE of the distribution of patterns lasting three beats and concluded that SE decreased during tilt due to the increased percentage of missing patterns. Dash et al. [116] described an algorithm for automatic detection of AF based on the randomness, variability, and complexity of the heartbeat interval time series. The algorithm combines three statistical techniques: root mean square of successive RR differences to quantify variability (RMSSD), SE to characterize its complexity, and turning points ratio (TPR) to test for randomness of the time series. The algorithm was tested on two databases, several sets of long RR intervals from many patients with and without AF and some with various forms of ectopic beats (MIT-BIH Atrial Fibrillation Database and the MIT-BIH Arrhythmia Database). The authors compared the sensitivity and specificity of the algorithm by varying the threshold of the SE from 0 to 1 at intervals of 0.01, and the optimal threshold was 0.7. The receiver operating characteristic (ROC) analyses were used to find data segment lengths. For the MIT-BIH Atrial Fibrillation Database, they achieved high sensitivity (94.4%) and specificity (95.1%). Considering MIT-BIH Arrhythmia, the algorithm has great performance even when tested against AF mixed with several other potentially confounding arrhythmias database (sensitivity = 90.2%, specificity = 91.2%). In Reference [117], 76 adults with persistent AF tested a novel application for the detection of an irregular pulse using an iPhone 4S. A smartphone application performed real-time pulse analysis using two statistical methods: square root of the successive RR difference (RMSSD/mean) and SE. The sensitivity, specificity, and predictive accuracy of both algorithms were examined using 12 electrocardiograms as the gold standard. The RMSSD/mean and SE were higher in participants in AF than in those with sinus rhythm. The authors showed that, for the discrimination of beat-to-beat of an irregular pulse during AF from sinus rhythm, the algorithm combining the two statistical methods demonstrated excellent sensitivity (0.96), specificity (0.98), and precision (0.97). In Reference [118], Lee et al. applied SE, RMSSD, and SampEn in data sets to discriminate between AF and normal sinus rhythm using an iPhone 4. The authors used 64-beat segments from MIT-BIH data sets, and they found the beat-to-beat accuracy value of 0.94, 0.93, and 0.96 for RMSSD, SE, and SampEn, respectively. They collected 2-minute pulsatile time series of 25 subjects with AF (pre and post-electrical cardioversion) with an iPhone 4S. Using threshold values derived from RMSSD, SE, and SampEn from the MIT-BIH data sets, they found beat-to-beat accuracy of 0.98, 0.85, and 0.95, respectively. Javorka et al. [119] tested if the complexity measures SE, MSE, and compression entropy provide diagnostic information regarding early subclinical autonomic dysfunction in diabetes mellitus. Seventeen patients with diabetes mellitus of type 1 were compared to a control group of 17 healthy subjects. The length of the RR intervals was measured for 1 h using a telemetric ECG system. The authors demonstrated that the magnitude and complexity of HRV are reduced in young patients with diabetes mellitus, indicating vagal dysfunction.

##### 3.8.2.2. Applications of Conditional Entropy and Corrected Conditional Entropy

Less than ten papers are published yearly using the CE measure in the heart-rate field. Porta et al. wrote the four most cited papers applying the CE to HR time series [101,120,121,122].

In Reference [101], which has been described in Section 3.8.2.1, Porta et al. show that CE increased during tilt and CR10 as patterns followed each other according to a more repetitive scheme and that, during CR10, SE and CE were not redundant as the regularity index significantly decreased while SE remained unchanged. In Reference [120], the measures CCE, ApEn, and SampEn were applied to the head tilt tests of 17 healthy nonsmoking humans (age 21 to 54 years old, 7 women). In this study, the subjects underwent, after 7 min at rest (R), a session (lasting 10 min) of head-up tilt (T) with table angles chosen within the set 15, 30, 45, 60, 75, 90 (T15, T30, T45, T60, T75, T90). An R session always preceded each T session and followed by 3 min of recovery. Each subject’s ECG was recorded and analyzed at all tilt angles but in random order. All measurements showed a progressive decrease as a function of the angles of inclination with a tendency to saturate at T75, thus indicating that complexity is under control of the autonomic nervous system. SampEn, ApEn, and CCE provide global indices that may be useful for monitoring the sympathovagal balance. A function was purposed to measure the degree of uncoupling between two signals in Reference [121]. This function exploits the ability of the CE calculated over two signals to give the amount of information carried by one of the signals when the samples of the other are known. The function was applied to quantify the degree of coupling between the heart period and ventricular repolarisation interval obtained from the surface ECG in human subjects (10 normal subjects and 10 myocardial infarction patients). In myocardial infarction patients uncoupling, the function was constant, thus indicating that the RR series are uncoupled. Reference [122] presented a cross-conditional entropy to assess causal relationships between heart period (HP) and systolic arterial pressure during graded head-up tilt. A traditional approach based on phases was applied for comparison. Porta et al. tested the ability of the approach to detect the lack of causal link from systolic arterial pressure to HP. It was assessed on 8 short-term and 11 long-term heart-transplant recipients and the spontaneous HP and systolic arterial pressure variabilities were extracted from 17 healthy humans (9 females) at rest and during graded head-up tilt (from $15{\;}^{\circ }$ to $75{\;}^{\circ }$ with steps of $15{\;}^{\circ }$). They detected the lack of causal relation from systolic arterial pressure to HP in short-term heart-transplant recipients and the gradual restoration of the causal link from systolic arterial pressure to HP with time after transplantation in the long-term heart-transplant recipients. The head-up tilt protocol induced the progressive shift from the prevalent causal direction from HR to systolic arterial pressure to the reverse causality with tilt table inclination in healthy subjects. Finally, Guzzetti et al. [123] evaluate whether the 24-h HRV spectral and nonlinear analysis, considered as autonomic cardiac modulation markers, contains independent prognostic information in patients with CHF. Periods of 300 RR intervals were analyzed from Holter recordings. Thirty patients with CHF and 20 control subjects were studied for 2 years. Patients were divided into survivors (22 patients) and non-survivors (8 patients) during the follow-up period. The power spectral analysis, the slope of the linear relationship between log-power versus log-frequency (1/f), and the complexity content using CCE of the RR series were calculated. The measure CCE was slightly lower, but not significantly, in patients than in controls (0.79±0.02 nats vs. 0.82±0.02 nats) and in survivors than in non-survivors (0.78±0.03 vs. 0.79±0.04).

##### 3.8.2.3. Applications of Approximate Entropy

Since ApEn was proposed, more than 280 papers have been published applying this method to the HR time series, creating more than seventeen thousand citations. The five most cited papers that applied ApEn to HR time series are Reference [19,64,103,124,125]. Pincus and Goldberger [103] gave the formal mathematical description of ApEn, applying it to two different clinical HR data sets. The authors discussed the algorithm implementation and interpretation and introduced a general mathematical hypothesis of the dynamics of a broad class of diseases, indicating the utility of ApEn to test this hypothesis. Lake et al. [124] applied the ApEn and SampEn to the study of 89 consecutive admissions to a tertiary care neonatal intensive care unit, among which there were 21 episodes of sepsis. They analyzed 4096 RR intervals collected during 25 min. The main results were that entropy values drop before clinical signs of neonatal sepsis. In Reference [19], described in Section 3.2.2, during the atropine administration, the changes in ApEn were variable and small with a trend toward lower values after the first dose and after a parasympathetic block observed a progressive increase of ApEn during exercise. Pincus et al. [125] applied ApEn temporal series of 1000 means of beat-to-beat HR (RR interval) from 24 infants being cared for in the Neonatal Special Care Unit (9 sick and 15 healthy). The ApEn values were calculated varying m from 1 to 3 and r from 1 to 5. They found that ApEn was higher for healthy neonates than for neonates sick with a significance for all pairs except for m=3 and r=1. In Reference [64], described in Section 3.5.2, the results showed that ApEn was not a predictor of survival after the diagnosis of CHF. The ApEn was slightly, but not significantly, lower in patients with CHF than in controls subjects (1.18±0.18 vs. 1.24±0.21). The authors thought these results are the consequence of the nonstationarity of the data sets and intrinsic limitations of the measure in the context of very low overall variability.

##### 3.8.2.4. Applications of Sample Entropy

Since SampEn was proposed, more than 250 papers have been published applying this method to HR time series. In Reference [124], described in Section 3.8.2.3, Lake et al. found that SampEn was significantly associated with upcoming sepsis (ROC area: 0.64, 95% confidence interval: 0.56;0.74, p=0.001). Also, SampEn significantly added diagnostic information to the variables gestational age and birth weight (p<0.001). Javorka et al. [126] checked the association between HR decrease after exercise and HRV parameters. The HR was monitored in 17 healthy male subjects during the pre-exercise (25 min supine, 5 min standing), exercise, and recovery phases (30 min supine). They calculated RR intervals mean, SDRR, RMSSD, the proportion of interval differences of successive RR intervals greater than 50 milliseconds (ms), log high and low frequency spectral powers, and SampEn. The rate of cardio-deceleration did not correlate with pre-exercise HRV parameters but positively correlated with HRV measures and SampEn obtained from the early phases of recovery. Heart-rate complexity was slightly reduced after exercise and attained rest values after 30-min recovery. Lake and Moorman [127] investigate the problem of AF detection in implanted ventricular devices. The authors optimized SampEn estimation and developed the algorithm called the sample entropy coefficient using the canonical MIT-BIH database (10-h recordings from 25 patients with AF with 1,221,578 intervals). They validated the algorithm in a new and much broader set of consecutive Holter monitors University of Virginia (1461 24-h RR interval from 940 Holters from patients (480 men) over 40 years old). In patients older than 40 years old, the SampEn and sample entropy coefficient presented high degrees of accuracy in distinguishing AF from normal sinus rhythm in 12-beat calculations performed hourly. Ahmad et al. [128] monitored the HR continuously in adult patients undergoing bone marrow transplantation (*n* = 21) starting one day before bone marrow transplant and continuing until recovery or withdrawal (1264 days). Fourteen patients developed sepsis requiring antibiotic therapy, whereas 3 did not. The SampEn and MSE were calculated in all the traces. On average, for 12 out of 14 infected patients, a significant (25%) reduction was observed in the SampEn and MSE before the clinical diagnosis and treatment of sepsis. For infected patients, wavelet HRV demonstrated a 25% drop from baseline 35 h before sepsis on average. For 3 out of 3 noninfected patients, SampEn showed a significant reduction. Finally, in Reference [120], described in Section 3.8.2.2, SampEn was calculated with m−1=2 and r=20% of the standard deviation. All parameters based on SampEn during T45, T60, and T75 were significantly smaller than those at rest.

##### 3.8.2.5. Applications of Multiscale Entropy

Since 2002, when the multiscale entropy (MSE) measure was proposed, more than 100 papers have been published applying this method to the HR time series, creating more than six thousand citations. The five most cited papers that applied MSE to HR time series are References [108,109,128,129]. Costa et al. [108] introduced the algorithm to calculate the MSE for complex time series using the SampEn for each coarse-grained time series plotted as a function of the scale factor. The authors applied MSE to compare the heart-rate time series of 20 elderly subjects, ten men and ten women (69±3 years), and to 20 healthy young subjects, ten men and ten women (32±6 years). They concluded that, for all time scales, a higher entropy value is attributed to the time series of young people, meaning a loss of HR complexity with age. Furthermore, the authors showed that MSE robustly separates healthy and pathologic groups and consistently yields higher values for simulated long-range correlated noise compared to uncorrelated noise. In Reference [109], Costa et al. described in detail the basis and implementation of MSE. The authors applied the method to RR time series derived from 24-h ECG recordings of 27 healthy young subjects (34.5±7.3 years, range 20–50 years), 45 healthy elderly subjects (70±3.97 years, range 66–75 years), 43 CHF subjects (aged 55±11.6 years, range 22–78 years) and 9 subject with AF. The method consistently indicates a loss of complexity with aging, with an erratic cardiac arrhythmia (atrial fibrillation) and with a life-threatening syndrome (CHF). Also, the authors compared the MSE values during sleep and wake periods for the different groups to assess the effects of activity level. The authors also applied the method to the analysis of coding and noncoding DNA sequences (four coding, nine noncoding DNA sequences from human chromosome 22 and 30 binary random time series). They found that the noncoding DNA sequences have a higher MSE. The third most cited paper of MSE measure [128] has been described in Section 3.8.2.4. In Referencer [129], MSE has been proposed to quantify the complexity of physical and physiologic time series. The authors applied the MSE method to the CinC 2002 test datasets, and the method correctly identified the origin (real and simulated RR interval time series) of 48 out of 50-time series. Valencia et al. [112] proposed a refined MSE in order to overcome the limitations of the MSE method: the artificial MSE reduction due to the coarse-graining procedure and the introduction of spurious MSE oscillations due to the suboptimal procedure for the elimination of the fast temporal scales. The authors applied refined MSE to 24-h Holter recordings of HRV obtained from 62 healthy subjects (41 men and 21 women, aged 39.8±11.2 years, range 21–64 years) and 148 subjects with aortic stenosis (86 men and 62 women, aged 60.6±9.8 years, range 26–81 years). In healthy subjects, during nighttime, refined MSE was larger at longer scales (*s* = 4–20) and smaller at short scales (s= 1–2) than during daytime. In aortic stenosis group, refined MSE was smaller during daytime both at short and long time scales (s= 1–11) than during nighttime. Refined MSE was larger in the healthy group than in the aortic stenosis group population during both day–time (s= 2–9) and nighttime (s= 2).

### 3.9. Symbolic Dynamics

#### 3.9.1. Description of Symbolic Dynamics

The concept of symbolic dynamics (SymD) goes back to Hadamard [130] and allows a simplified description of the dynamics of a system with a limited amount of symbols. For HRV analysis, the underlying theoretical concept is used in a rather pragmatic way. The main idea is to encode, according to some transformation rules, RR intervals and their changes. The authors applied MSE method to the CinC 2002 test datasets, and the method correctly identified the origin of 48 out of 50 time seriesfew symbols of a specific alphabet. Subsequently, the dynamics of that symbol string are quantified, providing more global information regarding HR dynamics. Two techniques introduced, by Voss et al. (1996) [131] and Porta et al. (2001) [101], are the most used ones.

##### Voss’s Technique

According to the SymD approach described by Voss et al., the series of RR intervals are transformed into an alphabet of 4 symbols—0, 1, 2, and 3—depending on how much single RR intervals differ from the mean. The transformation rules proposed are presented in the next equation, where μ is the mean of RR intervals and α is a particular scaling parameter (usually equal to 0.1). In this transformation, the symbols “0” and “2” indicate a small difference and “1” and “3” encode a large difference from the mean.
(32)$$\begin{array}{cc} \; & 0:\mu <x(i)\leq (1+\alpha )\cdot \mu \end{array}$$
(33)$$\begin{array}{cc} \; & 1:(1+\alpha )\cdot \mu <x(i)<\infty \end{array}$$
(34)$$\begin{array}{cc} \; & 2:(1-\alpha )\cdot \mu <x(i)\leq \mu \end{array}$$
(35)$$\begin{array}{cc} \; & 3:0<x(i)\leq (1-\alpha )\cdot \mu \end{array}$$

This method studies the probability distribution of words with three successive symbols from the alphabet to characterize symbol strings. In this way, one obtains 64 ($4^{3}$) different word types (bins). Several parameters characterize symbolic strings, such as the following:forbidden words (FORBWORD): the number of word types that occur with a probability less than 0.001; a high number of forbidden words reflect a reduced dynamic behavior in time series and vice versa.measures of complexity:−Shannon entropy—SE computed over all word types: a measure of word-type distribution complexity;−Rényi entropy (RE) *q* = 0.25—RE with a weighting coefficient of 0.25 computed over all word-types, predominately assessing the words with low probability;−Rényi entropy *q* = 4—RE with a weighting coefficient of 4 computed over all word-types, predominantly assessing words with high probabilities.wpsum—wpsum02 is measured as the percentage of words consisting of the symbols “0” and “2” only, and the wpsum13 is the percentage of words containing only the symbols “1” and “3”. According to the meaning of the symbols, high values for wpsum02 indicate low complexity of HR time series while high wpsum13 indicates higher complexity.

Note that the Rényi entropy, a generalization of the SE, is a family of functions of order q ($R_{q}$) defined as follows:(36)$$R_{q}=\frac{1}{1-q}ln\sum_{i}{p(x(i))}^{q}$$
where p(x(i)) is the probability of X=x(i). For the particular case when q=1, the SE is obtained.

Voss et al. also developed a modified approached of SymD for low or high variability. In this approach, the RR intervals time series is transformed in a symbolic string using a simplified alphabet consisting only of symbols “0” or “1”, where the symbol “0” stands for a difference between two successive beats lower than a particular limit and the symbol “1” represents those cases where the difference exceeds this particular limit. As time limits, 2, 5, 10, 20, 50, and 100 ms have been proposed. However, the limit 10 ms is the most used one since it is most useful according to hierarchical cluster and stepwise discriminant function analysis [132]. The low variability parameter is measured as the probability of occurrence of sequences containing six consecutive marks of “0” (plvar), whereas the high variability parameter (phvar) is calculated as the probability of sequences of six consecutive marks of “1”. Taken together, in this model, an increase of “000000” sequences, resulting in increased values of plvar, and a decrease in “111111” sequences, leading to reduced values of phvar, indicate reduced system complexity.

##### Porta’s Technique

In the symbolic analysis according to the approach described in Porta et al. (2001) [101], the subseries $X_{m}^{\xi }\left(i\right)=\left(X^{\xi }\left(i\right),X^{\xi }\left(i-1\right),\ldots ,X^{\xi }\left(i-m+1\right)\right)$ with i=m,…,N, previously defined in Section 3.8, are used to quantify patterns variation. The values of *m* and ξ have to be small in order to avoid a large number of possible patterns: for applications over short data sequences (250–300 samples), the best compromise has been shown to be ξ=6 and m=3 (216 possible patterns) [101]. To reduce the number of patterns without losing information, all the patterns were grouped into four families according to the number and types of variations from one symbol to the next one. The pattern families were as follows:patterns with no variation (0V, all the symbols were equal);patterns with one variation (1V, two consecutive symbols were equal, and the remaining one is different);patterns with two like variations (2LV, the three symbols formed an ascending or descending ramp);patterns with two unlike variations (2UV, the three symbols formed a peak or a valley).

The indexes 0V%, 1V%, 2LV%, and 2UV% are computed as the percentages of occurrence (number of times that a pattern $X_{m}^{\xi }\left(i\right)$ belonging to a specific family was found and divided by N−m+1 (multiplied by 100) of these families. Since the sum of all symbolic parameters is equal to 100% (i.e., 0V% + 1V% + 2LV% + 2UV% =100%), 0V% and 2UV% can increase or decrease at the expense of 1V% and 2LV% [133].

#### 3.9.2. Applications of Symbolic Dynamics

Hundreds of papers have been published applying the measure SymD to HR time series, creating more than four thousand citations. Voss et al. in 1996 [131] introduced a SymD method in a paper comparing the RR intervals of 61 subjects: 35 healthy persons, 10 patients after myocardial infarction with low electrical risk, and 16 cardiac patients after myocardial infarction high-risk therefore receive automatic defibrillators. They show advantages in combining several HRV methods, improving the precision of high-risk stratification. The same group compared, in a review paper, several nonlinear methods: DFA, ApEn, SampEn, MSE, compression entropy, SymD, and Pplot in Reference [8]. Effects of aging and gender differences were seen using the SymD (higher values were found in young vs. older males; higher values in older females vs. older males). Subjects with dilated cardiomyopathy (DCM), ischaemic heart failure (IHF), and myocardial infarction were also compared with a group of healthy controls. Significant higher values in the healthy group were found when applying the CE, SymD, and Pplot. Also, Voss et al. [134] computed several nonlinear indices, including the SymD, to determine age and gender reference values for representative 1906 healthy subjects. Wessel [135] aim for short-term prediction of early signs of ventricular tachycardia or ventricular fibrillation in 17 patients with an implanted cardioverter-defibrillator. They applied Voss’ method, obtaining a Shannon entropy of the word distribution significantly higher in the control group, whereas the short-variability measure “plvar10” is higher in the pathologic group. Both results indicate a decrease in heart-rate variability in the pathologic group. Maestri et al. [136] compared 20 nonlinear HRV indices, belonging to six major families: SymD, entropy, fractality-multifractality (DFA, FD-Higuchi’s, and HE), predictability, empirical mode decomposition, and Pplot in patients with CHF. As expected, variables within HVR indices from the same families tended to be highly correlated. However, high correlations were also observed between the four nonlinear variables of Pplot, predictability, empirical mode decomposition, and the HRV linear variables.

## 4. Discussion

The approaches to HRV analysis summarized in this paper contributed to the technical understanding of the signal character of the RR interval time series. HRV analysis is widely used to characterize the functions of the autonomic nervous system. However, caution should be exercised when using HRV indices because it is difficult to separate the heart-rate effect from the impact of another independent factor on HRV [137].

Malik and Camm [138] concluded that reduced HRV and decreased HRV components only indicate that periodic physiological fluctuations of the autonomic nervous system are minimized or not present. However, changes in the autonomic nervous system can be reduced by several factors (e.g., interventions, and pathologies). Therefore, the decrease in HRV cannot be directly interpreted as a specific change in the degree of activity and autonomic tone. In Reference [139], Sassi et al. states that the success of new approaches to HRV analysis in developing new clinical tools, such as those for identifying high-risk patients, has been somewhat limited. The authors note the need for multidisciplinary dialogue and specialized courses in combining clinical cardiology and complex signal processing methods for further advances in cardiac physiology and understanding of normal and abnormal cardiac control processes.

Methods relating the HR time series with other physiological time series, such as respiration and blood pressure, might be of great interest in clinical applications. Though, these methods, such as mutual information, cross-entropy, and Hausdorff distance almost always based on the techniques explored in this paper, should be carefully employed. Not only do they have the same limitations as they present others, e.g., related to the data synchronization but also the analysis of the outputs is not always trivial neither its clinical interpretation.

Nevertheless, there are many successful results using HRV analysis, as previously described. In Table 1, the applications/goals of the selected papers are displayed. Many of the most cited articles compare more than one nonlinear measure (e.g., [39,40,136]) or have more than one purpose, (e.g., References [8,54]). The main goals of the reviewed articles are to study (1) cardiac pathologies, such as AF and CHF; (2) the effect of age and gender; (3) the differences between night and day (or through the night); and (4) the HR dynamics before, during, and after exercise.

We notice that many studies use PhysioBank databases [13]. The increased number of (public) available datasets has contributed to an increase in published papers in this field, an essential progress in the clinical research area.

The vast number of articles using these methods made unfeasible the systematic review. Future work should try to understand which features of the traces each method is measuring and can be captured in simulated and real-world data.

The methods presented in this article have some limitations. The principal limitation is that most algorithms are parametric, mainly because the correct choice of the parameters is not always trivial and it will have an extreme influence on the results of the indices. Furthermore, we notice that many papers present neither the values of the parameters used nor a reason for the choice made. This problem increases when the many programs available to compute these indices have set default values for the required parameters, allowing the less experienced users to run the algorithms without questioning this problem. Table 2 describes some limitations of nonlinear methods for assessing heart-rate dynamics described in this article.

Many other measures are gaining more importance in the analysis of HRV, such as, Fuzzy entropy [140], permutation entropy [141], information storage [6], visibility graphs [142,143,144], compression [145,146], power spectral methods, and Petrosin’s fractal dimension methods [147]. These measures try to overcome some of the stated limitations of the methods presented in the paper. However, for many reasons, such as reduced number of citations, these measures were not covered in this article.

Moreover, as stated before, the main difficulty of using these methods in clinical practice is the difficult (or completely wrong) interpretation of the results. Many articles compared the results of these and other methods and probe their relationship with the more traditional linear heart-rate variability indices. However, a clear explanation of why some ways work better to distinguish some pathologies or in some specific data is not given. Additionally, questions of signal quality, frequency, and duration in real-world data should be explored in more detail.

## 5. Conclusions

This review paper reviews most common nonlinear methods applied to heart-rate time series. We enhance the review by introducing their most notable applications/results.

We found that the Poincaré plot has been one of the most used methods in the last years. It is a method easy to compute, it gives a visual display, and its interpretation is understandable by many clinicians. However, when trying to quantify the visual information, it presents some limitations such as it assumes an elliptical shape of the data distribution. On the other hand, while the use of entropies and DFA have become more frequent, the employment of fractal dimension in the heart-rate time series decreased. Also, the low number of papers using the Hurst exponent is associated with its relationship with the DFA algorithm.

The main contributions of this paper are as follows:a detailed description of the nonlinear methods most used to assess heart-rate dynamics presenting their relationship and stating the limitations of methods;a synopsis the most cited articles applying each measure to understand the applicability of the methods.

## Figures and Tables

**Figure 1 entropy-22-00309-f001:**
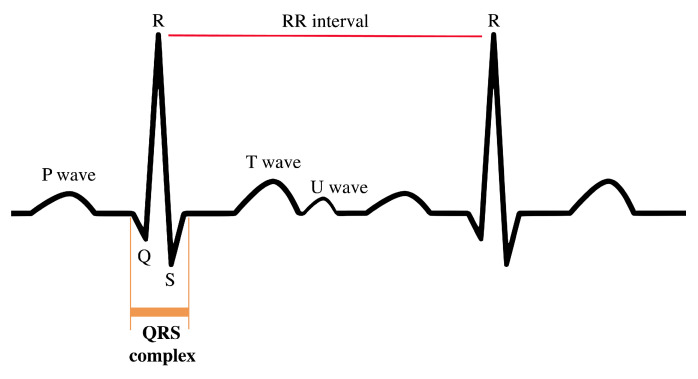
Schematic diagram of normal sinus rhythm for a human heart. Source: own elaboration.

**Figure 2 entropy-22-00309-f002:**
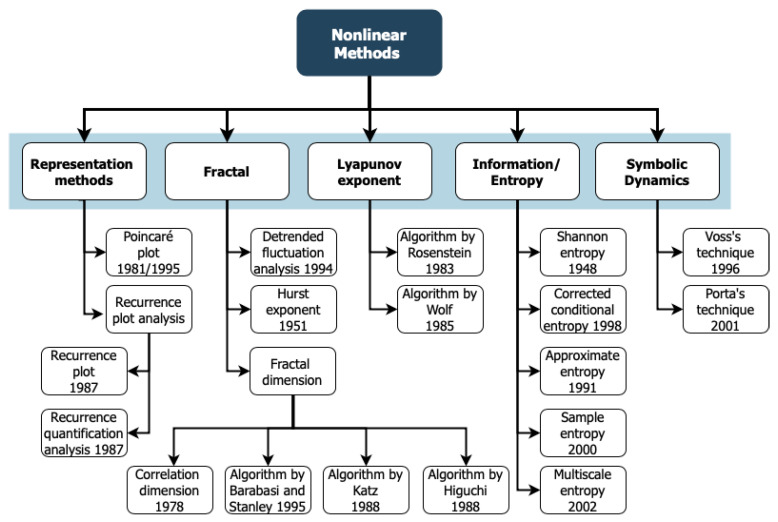
Nonlinear methods, with applications to heart-rate time series, presented in the paper.

**Figure 3 entropy-22-00309-f003:**
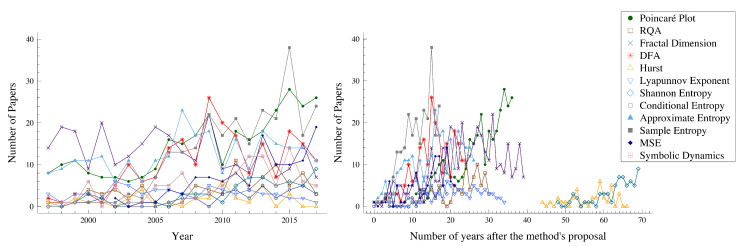
Number of published papers applying each method (described in Section 3) to heart-rate time series: Left panel—number of published papers by calendar year from 1997 to 2017, right panel—number of published papers by number of years after the method’s proposal. RQA—recurrence quantification analysis; DFA—detrended fluctuation analysis.

**Figure 4 entropy-22-00309-f004:**
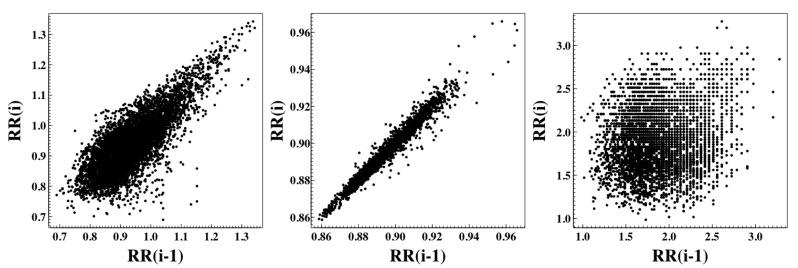
Poincaré plot for three RR time series: The left panel represents a normal sinus rhythm, the middle panel represents data from a congestive heart failure (CHF) patient, and the right panel represents the atrial fibrillation (AF) case. Note that the axes’ values are different in the three cases. Source: own elaboration.

**Table 1 entropy-22-00309-t001:** Applications of the nonlinear methods of the most cited papers selected.

Applications	Pplot	RP	FD	CD	DFA	HE	LE	SE	CCE	ApEn	SampEn	MSE	SymD
Theoretical application	[20,27]	[42]										[129]	
Aging	[8,86,134]		[54]	[54]	[8,54,63,66,86,134]		[54,86]			[8,54,66,86]	[8]	[108,109,129]	[8,134]
Gender	[8,134]		[54]	[54]	[8,54,134]		[54]			[8,54]	[8]		[8,134]
Physical Activity	[19,28]		[56]			[80]				[19]	[126]		
Orthostatic test	[26]												
Head-tilt test								[120]	[120,122]	[120]	[120]		
Stress	[39]	[39,41]		[39]	[39]					[39]			
Sleep			[57,81]	[81]	[65]	[81]	[81]			[81]		[109]	
Diabetes								[119]					
Sepsis											[128]		
Epilepsy		[42]											
Infants										[103,124,125]	[124]		
Cardiac Pathologies													
CAD	[40,78]	[40]			[40]	[78]		[40]	[40]	[40]	[40]		
Heart failure	[136]		[58,136]		[62,64,136]	[79,136]			[123]	[64]	[136]	[109]	[135,136]
AF		[43]						[116,117,118]			[118,127]	[109]	
Arrhythmia			[55]	[55]			[55]						[131]
Other	[8]			[49]	[8]	[49,77,80]	[49]		[121,122]	[8]	[8]	[112]	[8]

CAD—coronary artery disease; AF—atrial fibrillation; Pplot—poincaré plot; RP—recurrence plot; FD—fractal dimension; CD—correlation dimension; DFA—detrended fluctuation analysis; HE—Hurst exponent; LE—Lyapunov exponent; SE—Shannon entropy; CCE—corrected conditional entropy; ApEn—approximate entropy; SampEn—sample entropy; MSE—multiscale entropy; SymD—symbolic dynamics.

**Table 2 entropy-22-00309-t002:** Limitations of nonlinear methods most applied to heart-rate time series.

NONLINEAR METHODS	LIMITATIONS
**Representation Methods**	visual display techniques; several techniques to quantify the information.
**Poincaré Plot**	
SD1 and SD2	ellipse-fitting technique; lack of temporal information; correlation on other time-domain measures [8].
**Recurrence plot analysis**	
Recurrence Plot	parametric: needs *r*, *m* and τ parameters.
RQA	many measures hard to interpret.
**Fractal**	the algorithms give a number regardless of whether the object is factal.
**DFA**	parametric: needs *r*, *m*, and τ parameters; requires a choice of $\alpha_{1}$ and $\alpha_{2}$ split; assumption that the same scaling pattern is present throughout the signal; expects large time series; is a monofractal method; normal-to-normal interbeat intervals are required and dependency on editing ectopic beats [8].
**Hurst exponent**	hard to compute
**Fractal Dimension**	
Correlation dimension	parametric: needs *r*, *m*, τ, and *k* parameters; assumes linearity and/or exponential.
Box-counting dimension	parametric: needs ϵ.
Algorithm by Katz	heavily dependent on the record length; highly sensitive to the amplitude of noise.
Algorithm by Higuchi	parametric: needs *m* and *r*; sensitive to the amplitude of noise.
**Lyapunov exponent**	reflect effective growth rates of infinitesimal uncertainties over an infinite duration. However, time series analysis is restricted to the analysis of finite-time series, and thus, it is difficult to determine Lyapunov exponents [148].
**Algorithm by Rosenstein**	parametric: needs *m* and τ.
**Algorithm by Wolf**	does not take advantage to all the data; appropriate selection of maximum and minimum.
**Information/entropy**	measures the sequential regularity of contiguous events.
**Shannon entropy**	the distribution is not known; it cannot be used to compare diversity distributions that have different levels of scale; it cannot be used to compare parts of diversity distributions to the whole [149].
**CCE**	recoded RR and ξ has to be pre-chosen.
**Approximate entropy**	parametric: needs *r* and *m* parameters; heavily dependent on the record length and uniformly lower than expected for short records; counts self-matches; stationary data is required; inherent bias exists; lacks relative consistency; evaluates regularity on one time scale; outliers (missed beat detections, artefacts) may affect the entropy values [8].
**Sample entropy**	parametric: needs *r* and *m* parameters; a global marker of irregularity that might not represent reliably the local behavior in the neighborhood of a specific pattern and blur nonlinear feature [150]; stationary data is required; higher pattern length requires an increased number of data points; evaluates regularity on one scale; outliers (missed beats, artefacts) may affect the entropy values [8].
**Multiscale entropy**	artificial reduction of multiscale entropy due to the coarse-grained procedure; introduction of simulated oscillations due to the elimination of rapid time scales; lack an analytical framework allowing their calculation for known dynamic processes; reduction of reliability when applied in short time series [111].
**Symbolic dynamics**	detailed information will be lost; outliers (ectopic beats and noise) influence symbol strings [8].
**Voss’s technique**	parametric: need α time limit choice.
**Porta’s technique**	parametric: needs *m* and ξ.

SD1—short-term standard deviation; SD2—long-term standard deviation; RQA—recurrence quantification analysis; DFA—detrended fluctuation analysis; CCE—corrected conditional entropy.

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
