# Peer review of "Nonlinear Methods Most Applied to Heart-Rate Time Series: A Review"

_entropy, 2020, doi:10.3390/e22030309_

Round 1

Reviewer 1 Report

The paper was improved by revision, but my comments were in large part left unaddressed. Another revision is required in order to better take into account the variety of methods for the analysis of HRV (especially, in the information domain).

Specific comments:

1) The expansion of the introduction is not sufficient, especially regarding the definitions of complexity and its characterization. The authors now mention only the concept of Shannon Entropy, but complexity is assessed typically through the concept of conditional entropy (see my comment below). This should be mentioned explicitly, providing references, e.g. Xiong et al (Physical Review E 2017, 95(6), 062114) and references therein.

2) The section 3.7 about entropies has not been improved: as it is it appears just as a list of methods, and as such is of little use for the reader. Again, it should be better noted that conditional entropy is a general concept that is related to complexity, and that Corrected Conditional entropy, ApEn, SampEn, are just estimates of the concept of conditional entropy based respectively on the binning, kernel and nearest neighbor estimators. Please read again my previous comment, and make reference to the paper of Xiong et al (Physical Review E 2017, 95(6), 062114) and references therein to organize the section.

3) Again, I have to recall the opportunity to include Information Storage in the entropy based measures: it is not true, as the Authors state in their answer, that it depends on two time series! Make again reference to my previous comment: “another entropy measure is gaining more and more importance in the analysis of HRV is the information storage (IS). IS is actually a measure of regularity of individual HRV time series; it is related to the CE, while the CE itself is a measure of complexity; this should be acknowledged clearly. IS should be also included as one of the measures for which to review the recent applications on HRV.” See Xiong et al (Physical Review E 2017, 95(6), 062114) for the definition of IS and look into the references therein to find papers in which it was defined and used before.

4) Also the description of multiscale entropy is poor and needs to be better contextualized in the recent literature in the field of HRV. Please refer again to my previous commeny: “finally, recent developments of the method of multiscale entropy should be also acknowledged (e.g, Valencia et al. [IEEE Transactions on Biomedical Engineering, 2009, 56.9: 2202-2213], Faes et al. [Complexity 2017; 2017:1768264; . Physical Review E, 2019, 99.3: 032115]  Humeau Heurtier [Entropy 2015, 17(5), 3110-3123.])”

5) Figure 2 is interesting and useful. Please add Information Storage and Multiscale entropy as a box in the “Entropy section”

Author Response

We much appreciate the review of our manuscript, entitled "Nonlinear methods most applied to heart rate time series: a review."  We acknowledge the reviewers for their valuable comments and suggestions. We have modified our manuscript to accommodate the suggestions and corrections accordingly. Below, we address each comment.

The reviewers' comments are given below in bold (Arial), followed by our responses in a standard (Arial 11) font.

Reviewer 1

Comments and Suggestions for Authors

The paper was improved by revision, but my comments were in large part left unaddressed. Another revision is required in order to better take into account the variety of methods for the analysis of HRV (especially, in the information domain).

Specific comments:

1) The expansion of the introduction is not sufficient, especially regarding the definitions of complexity and its characterization. The authors now mention only the concept of Shannon Entropy, but complexity is assessed typically through the concept of conditional entropy (see my comment below). This should be mentioned explicitly, providing references, e.g. Xiong et al (Physical Review E 201795(6), 062114) and references therein.

The introductory paragraph was modified to reply to the reviewer's comment.

2) The section 3.7 about entropies has not been improved: as it is it appears just as a list of methods, and as such is of little use for the reader. Again, it should be better noted that conditional entropy is a general concept that is related to complexity, and that Corrected Conditional entropy, ApEn, SampEn, are just estimates of the concept of conditional entropy based respectively on the binning, kernel and nearest neighbor estimators. Please read again my previous comment, and make reference to the paper of Xiong et al (Physical Review E 201795(6), 062114) and references therein to organize the section.

We appreciate the comment. It is essential and, unfortunately, missed in all the changes we had made from the previous version. The conditional entropy and the corrected conditional entropy sections were separated. A new paragraph explaining the relation between the conditional entropy and its estimators was added.

3) Again, I have to recall the opportunity to include Information Storage in the entropy based measures: it is not true, as the Authors state in their answer, that it depends on two time series! Make again reference to my previous comment: “another entropy measure is gaining more and more importance in the analysis of HRV is the information storage (IS). IS is actually a measure of regularity of individual HRV time series; it is related to the CE, while the CE itself is a measure of complexity; this should be acknowledged clearly. IS should be also included as one of the measures for which to review the recent applications on HRV.” See Xiong et al (Physical Review E 201795(6), 062114) for the definition of IS and look into the references therein to find papers in which it was defined and used before.

We welcome the suggestion. We did the same procedure as for the other measures, and unfortunately, only a few papers appeared in our PubMed search. Several other entropy estimators as Fuzzy, permutation entropies were also not included in this paper and presented more references in the area. There are more than forty entropy measures applied to this area. We add a paragraph in discussion with reference to some of these measures. Sadly, we cannot include all in this paper.

4) Also the description of multiscale entropy is poor and needs to be better contextualized in the recent literature in the field of HRV. Please refer again to my previous commeny: “finally, recent developments of the method of multiscale entropy should be also acknowledged (e.g, Valencia et al. [IEEE Transactions on Biomedical Engineering, 2009, 56.9: 2202-2213], Faes et al. [Complexity 2017; 2017:1768264; . Physical Review E, 2019, 99.3: 032115]  Humeau Heurtier [Entropy 2015, 17(5), 3110-3123.])”

The multiscale entropy was not subject to our analysis for itself. The multiscale comprises two steps. The first one is the construction of the coarse-grained time series scales, and the second one applied an entropy measure to the scales. Many approaches were studied, both changing the first and second parts. As suggested, a discussion of this technique was refined in the discussion section.

5) Figure 2 is interesting and useful. Please add Information Storage and Multiscale entropy as a box in the “Entropy section”

We thank the reviewer for the comment. Please see the previous comments related to this topic.

Reviewer 2 Report

The article is still a technical review of the commonly used measures of heart rate variability that quantify the nonlinear properties of RR intervals. However this time the message is  clearly stated.

Again the comparisons of results of methods are still poorly discussed: Approximate Entropy vs Sample Entropy. 

The formula 24 must  explain p(x(i)) . 

Minor remarks:

The description of  plots in figure 3 should be increased for better readability.

In line 548 - threshold for SE - what is it?

In line 553 the bibliography item is not given.

The bibliography is untidy: capital letters usage. 

Author Response

We much appreciate the review of our manuscript, entitled "Nonlinear methods most applied to heart rate time series: a review."  We acknowledge the reviewers for their valuable comments and suggestions. We have modified our manuscript to accommodate the suggestions and corrections accordingly. Below, we address each comment.

The reviewers' comments are given below in bold (Arial), followed by our responses in a standard (Arial 11) font.

Reviewer 2

Comments and Suggestions for Authors

The article is still a technical review of the commonly used measures of heart rate variability that quantify the nonlinear properties of RR intervals. However, this time the message is clearly stated.

Again the comparisons of results of methods are still poorly discussed: Approximate Entropy vs Sample Entropy. 

We add a paragraph after the conditional entropy referring to the differences in binning estimation. The introduction of SampEn was slightly changed. And also, a new paragraph was added in the discussion section.

The formula 24 must explain p(x(i)).

We added the sentence: “p(x(i)) represents the probability of the point x(i).”

Minor remarks:

The description of plots in figure 3 should be increased for better readability.

We refined the figure 3 readability.

In line 548 - threshold for SE - what is it?

The threshold for SE means the value of SE chosen to balance the sensitivity and specificity of the algorithm.

In line 553 the bibliography item is not given.

The sentence was changed to incorporate the reference.

The bibliography is untidy: capital letters usage. 

We thank the reviewer for the notice. The references were adjusted accordingly.

Round 2

Reviewer 1 Report

This reviewer understands that it is not possible to exhaustively cover all aspects and algorithms of a wide field like nonlinear HRV analysis. However, achieving more balance among the different categories of methods would have been appropriate. Such a balance would have helped to gain attraction and citations for this work; for instance, limiting in Fig. 2 the entropy measures to conditional-entropy (complexity) based, without reporting regularity measures (e.g. information storage) and most importantly multiscale measures (e.g., MSE and its variants) makes such figure less comprehensive and impacting than it might have been.

Author Response

We much appreciate the review of our manuscript. We sincerely thank the valuable comment/suggestion. Taking that into consideration, we introduced the multiscale entropy as a measure: including the method description and its applications. Consequently, the discussion section, the figures, and tables were updated.